# The Reparative Function of MMP13 in Tertiary Reactionary Dentinogenesis after Tooth Injury

**DOI:** 10.3390/ijms25020875

**Published:** 2024-01-10

**Authors:** Henry F. Duncan, Yoshifumi Kobayashi, Yukako Yamauchi, Emi Shimizu

**Affiliations:** 1Division of Restorative Dentistry & Periodontology, Dublin Dental University Hospital, Trinity College Dublin, Lincoln Place, D02 F859 Dublin, Ireland; yamauchy@tcd.ie; 2Department of Oral Biology, Rutgers School of Dental Medicine, Newark, NJ 07193, USA; kobayayo@rwjms.rutgers.edu

**Keywords:** dentine, collagenase, matrix metalloproteinase, dentinogenesis, odontoblast, dental pulp

## Abstract

MMP13 gene expression increases up to 2000-fold in mineralizing dental pulp cells (DPCs), with research previously demonstrating that global MMP13 deletion resulted in critical alterations in the dentine phenotype, affecting dentine–tubule regularity, the odontoblast palisade, and significantly reducing the dentine volume. Global MMP13-KO and wild-type mice of a range of ages had their molar teeth injured to stimulate reactionary tertiary dentinogenesis. The response was measured qualitatively and quantitatively using histology, immunohistochemistry, micro-CT, and qRT-PCR in order to assess changes in the nature and volume of dentine deposited as well as mechanistic links. MMP13 loss affected the reactionary tertiary dentine quality and volume after cuspal injury and reduced Nestin expression in a non-exposure injury model, as well as mechanistic links between MMP13 and the Wnt-responsive gene Axin2. Acute pulpal injury and pulp exposure to oral fluids in mice teeth showed upregulation of the MMP13 in vivo, with an increase in the gene expression of *Mmp8*, *Mmp9*, and *Mmp13* evident. These results indicate that MMP13 is involved in tertiary reactionary dentine formation after tooth injury in vivo, potentially acting as a key molecule in the dental pulp during dentine–pulp repair processes.

## 1. Introduction

Dental pulp is a dynamic connective tissue encased by a mineralized shell of dentine and enamel. It has a formative role in tooth development, and a protective role by acting as a biosensor and secreting (via the odontoblast) new tertiary dentine after challenge by caries, trauma, and microbial leakage [1]. If the microbial threat is not managed, the pulp’s defenses can be overwhelmed, leading to a progressively more severe pulpitis, odontoblast death, and eventually, pulp necrosis [2]. However, if the inflammation is controlled and a sealing restoration is placed, the pulp can recover and repair [3]. The prevention and control of pulpitis forms the basis of operative dentistry; however, the difficulties in predictably treating deep caries have been highlighted in a recent international position statement [4], which concluded, “the control of pulpitis and conservative management of the exposed pulp should be a future focus of research activity” to reduce the provision of root canal treatment (RCT) and promote biomimetic restorative solutions [4,5]. Although RCT preserves the tooth, complete removal of the pulp is a destructive, technically difficult process [4,6], which removes inherent attributes of the pulp, including proprioception and immunity/repair, while increasing fracture and tooth loss [7,8]. There is an urgent need to understand the ‘key’ mediators that control pulpal repair and tertiary dentinogenesis formation in order to develop ‘smarter’ vital pulp treatments (VPTs) and immunotherapies, with the ultimate goal of creating new biologically based strategies aimed at modulating pulpitis [9,10], and therapeutically targeting tertiary dentine repair and regeneration [11,12].

Previous research has shown that hydraulic calcium silicate cements (HCSCs) effectively promote reparative tertiary dentine formation [13]; however, the mechanism is unclear [14]. Understanding the biology of the injured dentine–pulp complex enables identification of critical molecular mediators that orchestrate tertiary dentine formation and the immunological response to bacteria. These will underpin next-generation approaches and maximize the translational clinical impact in VPT [15]. Ideally, molecular mediators in dentine–pulp repair must be studied with and without inflammation, as this not only represents clinical reality, but also addresses the balance between inflammation and repair [5,16]. In this paper, specific molecular mechanisms controlling odontoblasts and tertiary dentinogenesis are studied using non-exposure and acute injury mouse models [17].

The current work focuses on a specific matrix metalloproteinase (MMP (MMP13)) that belongs to a family of host-derived, zinc-dependent endopeptidases [18] that degrade and reorganize extracellular matrix components [19], which we have recently shown to have critical functions in dentine–pulp tissues [20]. Furthermore, MMPs orchestrate multiple important dental tissue processes, including angiogenesis, differentiation, and chemotaxis [21,22]. Several in vitro and in vivo studies have investigated the influence of selected MMPs on enamel [23] and dentine formation [20], as well as the regulation of pulpitis [24,25,26]. The deletion of specific MMPs has resulted in an altered dentine structure in MMP9-knockout (KO) mice [16] and critical alterations to dentine after MMP13-KO in developing teeth, as shown by our research team [20].

MMP13 (collagenase-3) orchestrates bone formation and remodeling [27,28], and is the primary MMP involved in cartilage degradation in osteoarthritis [29]. As a result, systemic administration of pharmacological MMP13 inhibitors has been extensively tested in arthritis [30]; however, potential clinical development has been obstructed by the side effects of high-dose systemic application [29].

Recently, a key role for MMP13 in mineralization in osteoblasts [31], dental pulp cells (DPC) [32], and dentine formation [20] has been highlighted. MMP13 expression is high in pulp tissue [20,31,32] and, for the first time, we showed that MMP13 deletion resulted in critical alterations in the dentine phenotype, dentine–tubule regularity, the odontoblast palisade, as well as significantly reducing the dentine volume [20].

This evidence demonstrates MMP13′s importance in pulpal repair processes in vitro [32] and shows mechanistic links between MMP13 and the Wnt-signaling pathway [20]. However, it also highlights an absence of rigorous in vivo data investigating the role of MMP13 in tertiary dentine formation after injury and during pulpitis and potential molecular pathways for its action, and it remains unanswered as to whether MMP13 plays a central role in regulating tertiary reactionary dentine deposition. The aim of this study was to investigate the influence and mechanisms of MMP13 on the modulation of tertiary dentinogenesis using non-exposure and acute injury mouse models. The specific objectives were to qualitatively and quantitively analyze the nature and volume of tertiary dentine deposition in mouse molars after cuspal injury, as well as to assess mechanistic links to the Wnt-signaling pathway.

## 2. Results

### 2.1. MMP13 Loss Reduces Reactionary Dentine Quality, Volume, and Density after Injury

Tooth grinding injury to the mesial cusp of the maxillary first molar (pulp intentionally not exposed) in MMP13-KO (*Mmp13*^−/−^) mice showed a decreased reactionary dentine formation compared with wild-type (WT) mice at 6 months (Figure 1C,D) and 1 year (Figure 1E,F). Non-injured control samples showed no reactionary dentine deposition and no cuspal wear that could have been attributed to poorer formed dentine in the MMP13-KO group [20] (Figure 1A,B). Micro-CT analysis of mandibular molars indicated a significant tertiary dentine volume reduction of 28%, with an 8.6% decrease in density (Table 1). This supports our previously published data in a non-injury model showing reactionary dentine formation at the worn (but not injured) cusp tips of WT, but not MMP13-KO mice [20]. The data are based on our own group’s research methodology that used 6 mice per genotype, at 3, 6, and 12 months, and a healing interval of 4 weeks [33].

### 2.2. MMP13 Deletion Reduces Expression of the Wnt-Responsive Marker Axin2 after Cuspal Injury In Vivo

The Wnt-responsive gene Axin2, which is associated with odontoblast activity [34], was downregulated in injured *Mmp13^−/−^* samples after cuspal grinding, but not pulp exposure at 3 months (Figure 2A,B). This supports previous links identified between MMP13 and Wnt signaling during tooth development in our previous research [20]. The gene was also measured qualitatively by IHC at 6 and 12 months, showing a reduction in the staining in the pulp tissue (nucleus and cytoplasm) during reactionary dentinogenesis after injury in MMP13-null mice (Figure 2C–F).

Furthermore, the Axin2 IHC stain was quantified in the pulp tissue after masking of the dentine and pre-dentine layers at 6 months, with a significant reduction in the staining density evident in MMP13-KO compared with WT samples (*p* = 0.025) when analyzed using the IHC image profiler (ImageJ) [35,36] (Figure 3).

### 2.3. MMP13 Loss Results in Reduced Expression of the Pre-Odontoblast Marker Nestin after Injury

The pre-odontoblast marker was also measured qualitatively and quantitatively analyzed by IHC, showing a reduction in the odontoblast layer during reactionary dentinogenesis after injury in MMP13-null mice (3 months, 6 months, and 1-year-old; Figure 4). Furthermore, the Nestin staining intensity was quantified in the odontoblast layer and processed after masking other areas of the tooth at 6 months (magnification ×40), with a significant reduction in the staining density evident in MMP13-KO compared with WT samples (*p* = 0.039) when analyzed using the IHC image profiler (ImageJ; Figure 5).

### 2.4. MMP13 Expression Is Increased in Acutely Injured Teeth in WT Mice In Vivo in Combination with Other MMPs (MMP8 and 9)

A range of MMPs have been shown to be upregulated in chronically inflamed pulp tissue [37]; however, the expression of MMP13 after acute injury and contamination is unknown. MMP13 expression is upregulated in vivo in WT mouse molars after acute injury and exposure to the oral environment for 6 h (Figure 6). In comparison, there was very little expression in the non-injured contralateral teeth (Figure 6B). This expression of MMP13 was greater in the area adjacent to the exposure (Figure 6C,D), suggesting the local upregulation of MMP13 expression focused on the site of injury (Figure 6A,B).

MMP13 IHC staining intensity was quantified in the pulp tissue adjacent to the acute injury and left exposed to oral fluids for 6 h and compared with contralateral non-injured first molars at 6 months. A significant reduction in the staining density in the dental pulp was evident in non-injured sites compared with acutely injured teeth samples (*p* = 0.01; IHC profiler, ImageJ; Figure 7).

Furthermore, in pulp tissue taken from the acutely injured teeth exposed to the oral environment for 24 h, MMP8, 9, and 13 were all significantly upregulated compared with the control (Figure 8). MMP9 was upregulated over 6-fold compared with WT controls.

## 3. Discussion

Understanding the biology of the injured dentine–pulp complex enables identification of critical molecular mediators that orchestrate tertiary dentine formation and the immunological response to bacterial challenge. These will underpin next-generation approaches and maximize the clinical impact in vital pulp treatment [15]. We have previously demonstrated that MMP13 is related to the expression of key odontoblast-regulating factors (Osterix and Nestin) in the developing dentine–pulp complex [20], so the purpose of the current study was to ascertain the effect of MMP13 on reparative processes after injury in the dentine–pulp complex. The use of animal injury models is of significant benefit in the development of novel targeted therapeutic solutions based around MMP13 and related pathways, as understanding their action in 2D culture is limited in its translational capacity. Indeed, it is essential that molecular mediators of dentinogenesis must ideally be studied in vivo, as this represents the clinical reality and addresses the critical balance between inflammation and repair [5,17]. In this study, we have demonstrated that MMP13 acts differently in pulp tissue than in other tissues [22], highlighting the unique environment of the dentine–pulp and the potential opportunity to topically activate MMP13 to stimulate repair and reduce inflammation, while avoiding side effects.

The current study highlights, for the first time, a decrease in reactionary dentine depositions in injured MMP13-KO molars, which supports previous developmental data demonstrating a role for MMPs in primary dentine formation, by using MMP9-KO [16] and MMP13-KO models [20]. Specific MMP13 research has highlighted alterations in dentine structure, volume, and density after MMP13 ablation [20]. MMP13 has previously been shown to have an essential role in osteogenesis [27,28], is highly expressed in pulp tissue [38], and further increased during mineralization in rodent DPC culture [32,39], with a >2000-fold increase in human DPC cultures [20]. This work focused on reparative processes, during which it was shown that MMP13, along with MMP8 [40] and 9, significantly increased in dental pulp tissue after acute injury and exposure to the oral environment for 6 and 24 h, compared with non-exposed contralateral teeth (Figure 6 and Figure 8). Using a mouse cuspal injury model, the response of the dentine–pulp complex to various degrees of injury and mouse age was assessed in the presence and absence of MMP13. Initially, a non-exposed model was employed in which the mesial cusp of the maxillary first molar was ‘ground’ with a bur to a standardized depth, which was used to stimulate reactionary dentinogenesis. Reactionary dentinogenesis involves the upregulation of existing odontoblast activity in a defensive process aimed at distancing the pulp from the irritating stimulus [1]. Although odontoblast death and reparative dentine formation may also contribute to the repair using this technique, the bulk of tertiary dentine formed will be reactionary in nature. Within this study, qualitatively and quantitatively the volume of dentine deposited in MMP13-KO molars from various ages of mice were less than in WT matched controls (Figure 1), with the reactionary dentine volume significantly reduced by 28% and the density by 8.6% in 3-month-old mice (Table 1). The reduction in quality and quantity of tertiary dentine suggests that MMP13 has a role in promoting odontoblast activity and tertiary dentine deposition.

As a result, it was important to first analyze the odontoblast activity and, thereafter, the possible mechanisms by which MMP13 may affect dentine secretion after injury. In order to assess odontoblast activity, expression of the pre-odontoblast marker Nestin was assessed in vivo by IHC in mice aged from 3 months to 1 year and was shown to be increased in expression in WT mice molars compared with their *Mmp13^−/−^* counterparts, a finding that was similar to that observed in developing teeth at a range of time points [20]. It would appear from this that MMP13 loss diminishes the expression of this key odontoblast marker (Figure 4 and Figure 5). In an attempt to demonstrate a mechanism for this, the role of Wnt signaling was investigated, with a specific focus on the relationship with Axin2 in the injured pulp tissue. Axin2 has been previously highlighted as an effective marker not only of Wnt signaling but also of odontoblast activity by our group [20] and others working in this area [41,42]. In the current study, the Wnt-responsive gene Axin2, which is associated with odontoblast activity, was downregulated in injured *Mmp13^−/−^* samples (Figure 2 and Figure 3).

In order to move this research translationally toward the clinic, an acute injury mouse model was also employed, in which the pulp tissue was exposed on one side of the mouth in a standardized manner and the pulp was left open to oral contaminants for 6 and 24 h. This allowed the observation of inflammation before pulp necrosis occurred, and it could be compared with unexposed and uninflamed pulp on the contralateral side. Classic markers of pulpitis, MMP9 [25] and MMP8 [37], were, as expected, significantly increased after pulp exposure and contamination with oral bacteria; however, MMP13 also showed a significant increase in expression (an approximately 1.6-fold increase) of a similar magnitude to MMP8, but not as great as MMP9 (an approximately 6-fold increase), which highlights a potential novel role for MMP13 in inflammation as well as repair within the dentine–pulp complex. These data highlight a potential and critical role for MMP13 in repair within the dentine–pulp complex. The use of a global knockout in this study provided robust data using a validated model [20,43]; however, in the future, the potential use of a conditional KO model will increase the relevance of the research but may reduce the effect on the phenotype, as other supporting cells in the dentine–pulp complex are not affected. While further experimental work will be required to validate the effect of MMP13 ablation related specifically to the odontoblast cell, this study has, nevertheless, confirmed that there is an important role for MMP13 in stimulating mineralization processes in the injured pulp and highlighted an important novel target for the development of next-generation vital pulp treatments [44]. This study is potentially limited by the fact that it did not analyze the comparative injury response after pulpal exposure, focusing only on the response to injury in the absence of exposure. Although reactionary dentinogenesis is of considerable importance, the response of the exposed pulp is also of great translational interest and will be the focus of our future work in this area.

## 4. Materials and Methods

### 4.1. Animals

Homozygous MMP13-deficient mice (*Mmp13^−/−^*) on a C57BL/6 background were gifted by Dr. J D’Armiento. Male WT and MMP13-KO mice were characterized, housed, and cared for, as described previously [20]. Experimental (*Mmp13^−/−^*) and WT control animals were euthanized at a range of time points. Animal weights were recorded at the beginning of experimentation and at later time points. All experiments followed protocols approved by Rutgers University Institutional Animal Care and Use Committee (IACUC). All animal work complied with the Animal Research: Reporting in Vivo Experiments (ARRIVE) guidelines. The numbers of animals used are documented in Appendix A.

### 4.2. Mouse Injury Models

Mice (6 months old) were anesthetized using ketamine HCl (62.5 mg/kg) and xylazine (12.5 mg/kg) in sterile PBS by intraperitoneal injection and then mounted on a jaw retraction board. For tooth injury without pulp exposure, the occlusal surfaces of the right maxillary and right mandibular first molars were injured by cuspal grinding (50% depth) using a No. 2 round bur drill and air-dried. For acute tooth injury with pulp exposure, the occlusal surfaces of the right maxillary first molars were injured until pulp was exposed, using a No. 2 round bur drill with water spray, followed by an endodontic size K6-file (Flexofile, Maillefer, Dentsply-Sirona) for final exposure, air-dried, and sealed with glass ionomer cement (Fuji II, GC company, Chicago, IL, USA), as previously described [33]. The right injured maxillae were used for histology, and the right injured mandibles were used for micro-computed tomography (CT) or gene expression analysis. The left side of the maxillae was used as a control (non-injured site) for histology or the analysis of gene expression. For tooth injury without pulp exposure, mouse heads were collected 4 weeks following tooth injury. For acute tooth injury with pulp exposure, 3- and 6-month-old mouse heads were collected after 6 or 24 h of contamination in oral fluids (6 mice per genotype for each age and contamination time point).

### 4.3. Histological Analysis

Mouse mandibles were dissected for micro-computed tomography (µCT) assessment, while the remainder of the head was fixed in 10% buffered formalin at 4 °C for 24 h, prior to decalcification in 10% EDTA. The fixed tissue was dehydrated through ascending concentrations of ethanol, paraffin-embedded, and serially sectioned (5 μm). Thereafter, comparative sections were deparaffinized, hydrated, and stained with hematoxylin and eosin (H&E; Sigma-Aldrich, Saint Louis, MO, USA) prior to morphological analysis at 3 months, 6 months, and 1 year using a Zeiss Axio (Carl Zeiss, Jena, Thuringia, Germany) light microscope.

### 4.4. Immunohistochemical (IHC) Analysis

To detect the expression and distribution of the relevant mineralization- and inflammation-associated markers: Nestin, Axin2, and collagenase-3 (MMP13), IHC was carried out in similar sections using the Envision+ Horseradish Peroxidase (HRP) staining system (Dako, Agilent, Cork, Ireland) according to the manufacturer’s instructions. Briefly, deparaffinized and hydrated sections were rinsed in PBS prior to endogenous peroxidase activity, being blocked with a 0.3% hydrogen peroxide solution for 20 min (Dual Endogenous Enzyme Block, Dako). Sections were rinsed and blocked using a 1% bovine serum albumin (BSA) solution with 2% goat’s serum (Santa Cruz Biotechnology, Heildelberg, Germany) and 0.05% Tween-20 (Bio-Rad, Hertfordshire, UK) for one hour at room temperature. Sections were incubated with a primary antibody in a TBS solution with 1% BSA overnight at 4 °C in a humidifying chamber. MMP13 expression was analyzed in WT mice using anti-MMP13 (Abcam, Cambridge, UK). Sections from *Mmp13^−/−^* and WT were incubated with anti-Nestin (Millipore, Merck Life Science, Arklow, Ireland), MMP13 (WT only, Abcam), and Axin2 (Abcam) antibodies, or with PBS replacing the primary antibodies as a negative control. Positive control was demonstrated by antibody expression in tissues previously shown to have high expression [20]. After washing, sections were placed in HRP-labeled polymer conjugated to goat anti-rabbit and anti-mouse secondary antibodies (Envision+, Dako) for 30 min, prior to completion of staining with a 5-min incubation with 3,3′-diaminobenzidine (DAB) chromogen solution, washing in distilled water, and counterstaining with hematoxylin (Sigma-Aldrich). As before, a Zeiss Axio (Carl Zeiss) light microscope was used. The expression of Axin2, Nestin, and MMP13 was scored and quantified according to the stain intensity using the IHC Profiler plugin in ImageJ (version 1.54g), as previously described [35] and modified [36].

### 4.5. Micro-Computed Tomography (µCT)

The mandibles of *Mmp13^−/−^* and WT mice were fixed in 70% ethanol and prepared for high-resolution µCT (SkyScan 1172, Bruker, Kontich, Belgium) of the first molar teeth at 3 months. A three-dimensional analysis was carried out to determine the total tertiary dentine volume and total mineral density (TMD). The samples were scanned using a 10-MP digital detector, 10 W of energy (70 kV and 142 mA), a pixel size of 7.5 microns, exposure of 850 ms/frame rotation step of 0.3 degrees with ×10 frame averaging, a 0.5 mm aluminum filter, and scan rotation of 180 degrees. After scanning, the radiographs were reconstructed using NRecon software (version 1.7.3.0; Bruker, Billerica, MA, USA) and GPU acceleration. Gaussian smoothing was applied with a 2-voxel radius. Ring artefact and beam-hardening corrections were applied in the reconstruction. Ring artefact reduction was set to 7 pixels. Beam-hardening correction was set to 40%. CTAn software (CTAn Micro-CT Software, version 1.18.4.0, Bruker) was used to generate 2D images for color density and 3D images for the CT volume.

### 4.6. Quantitative Real-Time PCR (qRT-PCR)

Total RNA from DPCs/pulp tissue was isolated using the TRIzol (Thermo Fisher Scientific, Lenexa, KS, USA) method, and reverse transcribed to complementary DNA (cDNA) with TaqMan Reverse Transcription Reagents (Thermo Fisher Scientific), as previously described [20]. The sequences were amplified by adding complementary DNA to the PCR mixture containing each primer (Table 2) and Platinum SYBR Green qPCR SuperMix uracil-DNA glycosylase (UDG; Thermo Fisher Scientific). The reactions were pre-incubated at 50 °C for 2 min for decontamination of deoxyuridine (dU)-containing DNA by UDG and then incubated at 95 °C for 2 min to inactivate UDG and activate Taq. The PCR program continued for 46 cycles of denaturation at 95 °C for 15 s, annealing at 60 °C for 30 s, and elongation at 72 °C for 30 s. All data were normalized by the Ct value of beta-actin gene expression from the same sample.

### 4.7. Statistical Analysis

Student’s *t*-test was used for quantitative analysis of µCT and qRT-PCR data. Results were expressed as mean ± standard deviation. Data analysis was performed using IBM SPSS (v25, Dublin, Ireland; *p* < 0.05). The number of independent experiments or animals is listed in the results and figures.

## 5. Conclusions

This manuscript presented robust in vivo evidence to support a role for MMP13 in the formation of high-quality reactionary dentine formation after injury. The loss of MMP13 affected both the dentine quality and volume, with a reduction in the expression of Nestin and Axin2. This highlighted an important role for MMP13 in the organizing and regulating dentine–pulp reparative processes. From a translational perspective, the results of the present study showed mechanistic links to Axin 2 and the β-Catenin-Wnt signaling during tertiary dentinogenesis, thereby identifying potential novel targets in dentine–pulp repair processes.

## Figures and Tables

**Figure 1 ijms-25-00875-f001:**
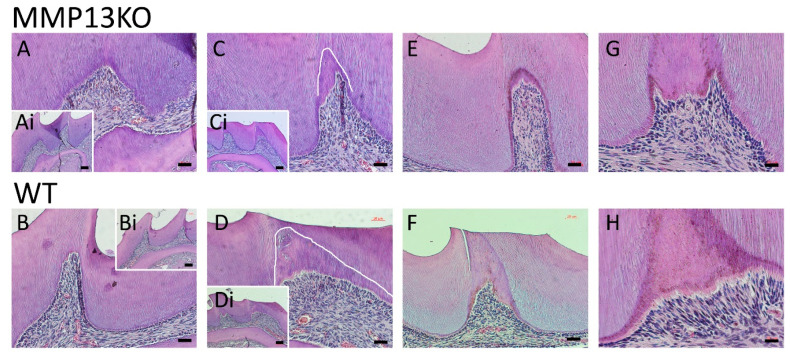
Comparative histological analysis of reactionary dentine formation and pulpal response in 3-month-, 6-month-, and 1-year-old *Mmp13*^−/−^ and WT mice after the mesial cusp of the maxillary right first molar was injured. (**A**) Mesial cusp of the uninjured (control) left maxillary first molar of 3-month-old *Mmp13*^−/−^ and (**B**) WT mice. A lack of reactionary dentine is evident in both samples. (**Ai**,**Bi**) Lower magnification view of (**A**,**B**) highlighting the uninjured cusp and cusp tip free from wear. (**C**) Mesial cusp of the injured right maxillary molar in 6-month-old *Mmp13*^−/−^ mice, showing a small deposition of reactionary dentine (outlined in white), while (**D**) WT mice demonstrated a significant volume of reactionary dentine with a tubular structure. (**Ci**,**Di**) Insets highlight an iatrogenic injury to the mesial cusp. Reactionary dentine response in different teeth at 6 months in *Mmp13*^−/−^ (**E**) and WT (**F**) mice, highlighting increased reactionary dentine formation in the WT sample, compared with minimum deposition in the knockout tooth (**G**). High magnification of the mesial cusp of an injured right maxillary molar in 1-year-old *Mmp13*^−/−^ and (**H**) WT mice highlighted a clear pre-dentine layer and increased odontoblast presence in WT. Scale bars = (**A**–**F**) 20 µm (original magnification ×20), (**Ai**–**Di**) 50 µm (original magnification ×10), and (**G**,**H**) 10 µm (original magnification ×40). *n* = 6 at each time point for both genotypes.

**Figure 2 ijms-25-00875-f002:**
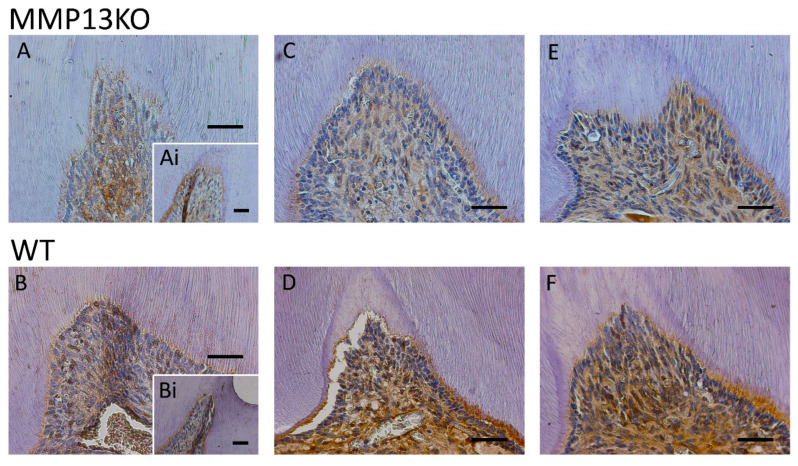
Comparative histological analysis of Axin2 expression during reactionary dentine formation in 3-month-, 6-month-, and 1-year-old *Mmp13^−/−^* and WT mice after the mesial cusp of the maxillary right first molar was injured. Comparative analysis of Axin2 expression (**A**,**B**). Mesial cusp of an injured right maxillary molar of 3-month-old (**A**) *Mmp13^−/−^* and (**B**) WT mice, showing increased expression in pulp tissue compared with the (**Ai**,**Bi**) uninjured (control) left side of each mouse. (**C**) Mesial cusp of an injured right maxillary molar in 6-month-old *Mmp13^−/−^* mice, showing decreased expression compared with (**D**) WT mice. (**E**) Mesial cusp of an injured right maxillary molar in 1-year-old *Mmp13^−/−^* mice, again showing decreased expression compared with (**F**) WT mice. All scale bars = 20 µm (original magnification ×40). *n* = 6 at each time point for both genotypes.

**Figure 3 ijms-25-00875-f003:**
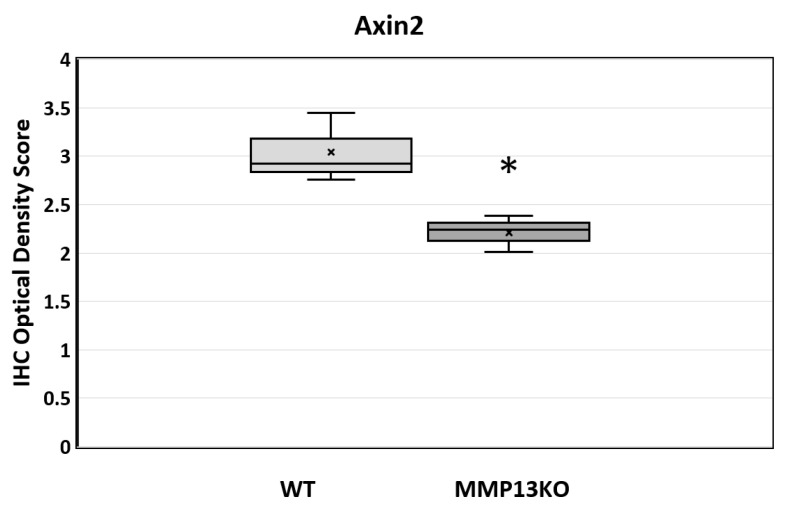
The quantitative analysis of Axin2 expression determined by the IHC staining optical density score in WT and MMP13-KO mouse pulp tissue at 6 months. The pulp tissue in the injured mesial pulp horn only was analyzed, and any pre-dentine staining was eliminated from the analysis to exclude potential artificial background factors. *n* = 6 for each genotype (ImageJ v 1.54g). Data represented by ‘x’ = median, horizontal line in box = mean, box = 25–75%, whisker = minimum/maximum. * *p* < 0.05 versus control.

**Figure 4 ijms-25-00875-f004:**
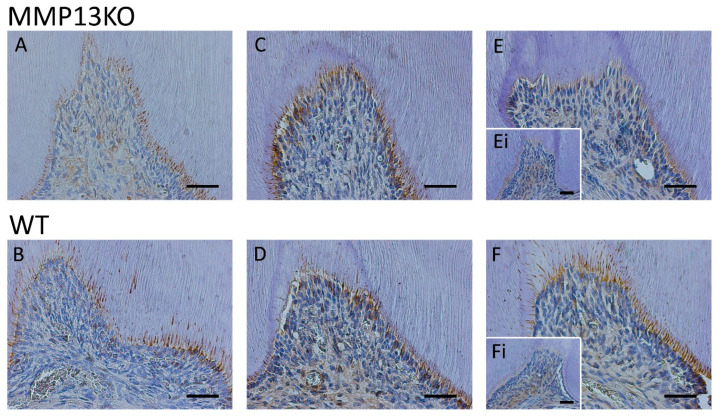
Comparative analysis of Nestin expression in maxillary first molars of 3-month-, 6-month-, and 1-year-old *Mmp13^−/−^* and WT mice after injury to the mesial cusp. (**A**) Mesial cusp of an injured right maxillary molar in 3-month-old *Mmp13^−/−^* and (**B**) WT mice, showing high expression in the pre-dentine layer. (**C**) Mesial cusp of an injured right maxillary molar in 6-month-old *Mmp13^−/−^* and (**D**) WT mice, showing higher expression in the pre-dentine layer of WT. (**E**) Mesial cusp of an injured right maxillary molar in 1-year old *Mmp13^−/−^* and (**F**) WT mice, showing increased expression in the pre-dentine layer compared with the (**Ei**,**Fi**) uninjured (control) left side of each mouse. All scale bars = 20 µm (original magnification ×40). *n* = 6 at each time point for both genotypes.

**Figure 5 ijms-25-00875-f005:**
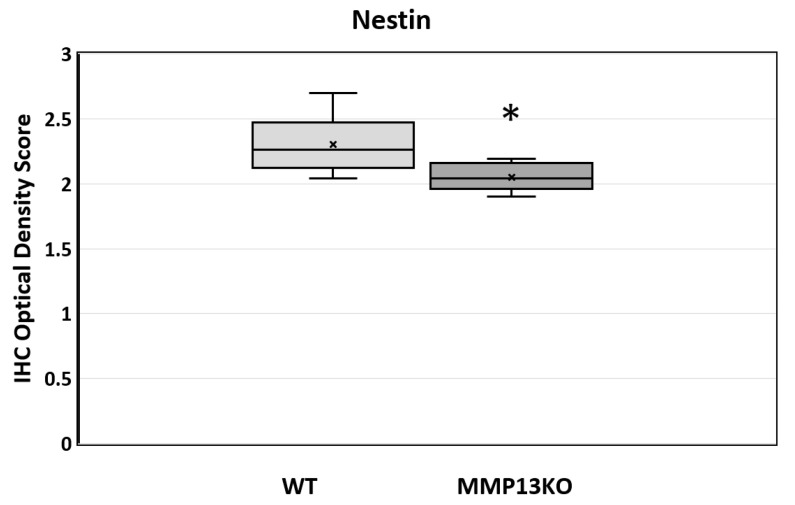
The quantitative analysis of Nestin expression determined by the IHC staining optical density score in WT and MMP13-KO mouse pulp tissue at 6 months. The tissue in the injured mesial cusp only was analyzed, and staining analysis was confined to the odontoblast layer and processed to exclude potential artificial factors. *n* = 5 for each genotype (ImageJ v 1.54g). Data represented by ‘x’ = median, horizontal line in box = mean, box = 25–75%, whisker = minimum/maximum. * *p* < 0.05 versus control.

**Figure 6 ijms-25-00875-f006:**
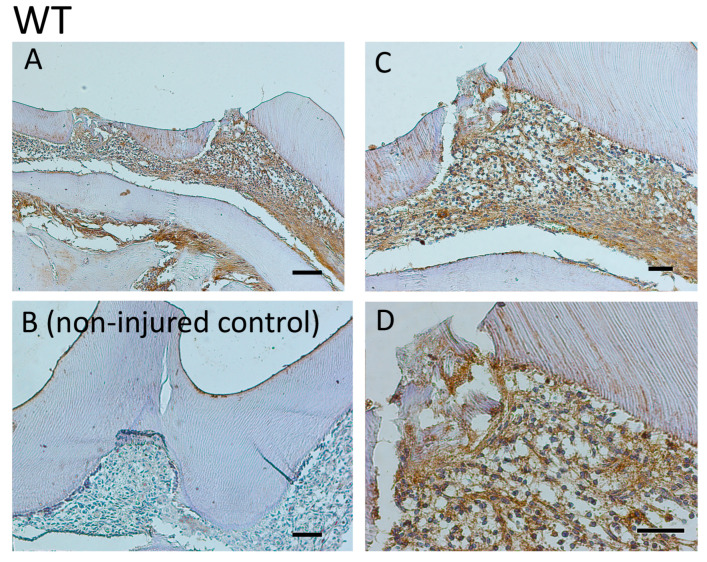
Comparative histological MMP13 expression analysis of injured 6-month-old WT maxillary first molars with pulpal exposure left open to the oral environment for 6 h. (**A**) MMP13 expression was high in the pulp tissue and alveolar bone of injured right molars compared with expression in non-injured left molars (**B**). (**C**) Higher magnification revealed intense MMP13 expression in pulp tissue and dentine. (**D**) Higher magnification of the exposed area in (**C**) revealed high expression in the pulp matrix, odontoblast, and around the exposure site. Scale bars = (**A**) 50 µm (original magnification ×10), (**B**,**C**) 20 µm (original magnification ×20), and (**D**) 20 µm (original magnification ×40). Sections were examined from 6 WT mice.

**Figure 7 ijms-25-00875-f007:**
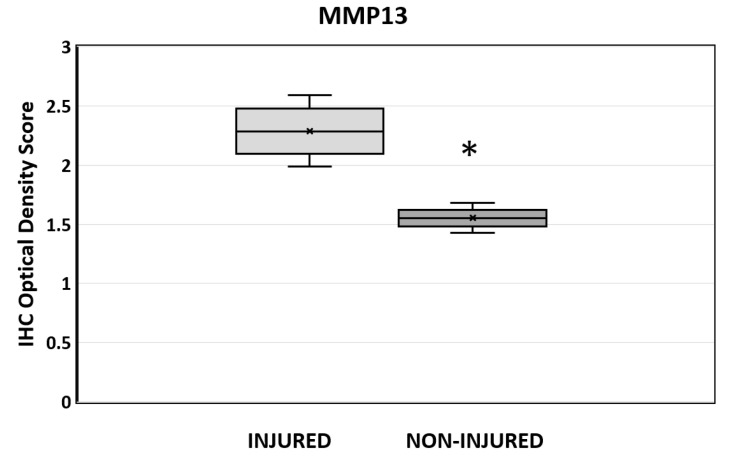
The quantitative analysis of MMP13 expression determined by the IHC staining optical density score in WT mice at 6 months from injured and non-injured sites. The pulp tissue adjacent to the acutely inured site was analyzed, and any pre-dentine staining was eliminated from the analysis to exclude potential artificial background factors. *n* = 6 for each group (injured and non-injured; ImageJ v 1.54g). Data represented by ‘x’ = median, horizontal line in box = average, box = 25–75%, whisker = minimum/maximum. * *p* < 0.05 versus control.

**Figure 8 ijms-25-00875-f008:**
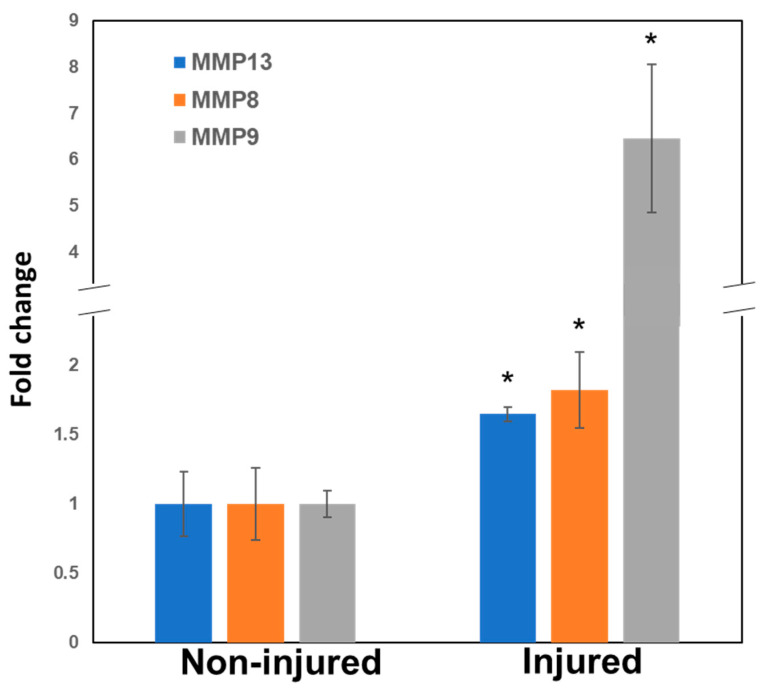
MMP8, 9, and 13 gene expressions were all significantly upregulated in pulp tissue from injured molars in 3-month-old WT mice after acute injury and exposure to oral contamination for 24 h. Data are shown as mean ± s.d., * *p* < 0.05 versus control. Data analyses were carried out in triplicate on three independent biological samples.

**Table 1 ijms-25-00875-t001:** Micro-CT (µCT) analysis of the mandibular first molar of 3-month-old WT compared with *Mmp13^−/−^* mice showed a significant decrease in the volume and density of reactionary dentine deposition after MMP13 ablation. ^a^ Averaged volume (mm^3^) and mineral density (g/cm^3^) from five samples. Values in parentheses are standard deviations. ^b^ The *p*-values are calculated using two-tailed Student’s *t*-test.

	Wild Type ^a^	MMP13 Knock-Out ^a^	*p*-Value ^b^	% Decrease
Volume	0.0567 (0.0036)	0.0409 (0.0060)	0.003	27.8
Density	1.5987 (0.0397)	1.4599 (0.0447)	0.008	8.6

**Table 2 ijms-25-00875-t002:** Primer sequences for qRT-PCR.

Gene	Primer Sequence (5′ to 3′)
Mouse MMP8	(F)-GATGCTACTACCACACTCCGTG (R)-TAAGCAGCCTGAAGACCGTTGG
Mouse MMP9	(F)-GCTGACTACGATAAGGACGGCA (R)-TAGTGGTGCAGGCAGAGTAGGA
Mouse MMP13	(F)-GATGACCTGTCTGAGGAAGACC (R)-GCATTTCTCGGAGCCTGTCAAC
Mouse beta-actin	(F)-TCCTCCTGAGCGCAAGTACTC (R)-CGGACTCATCGTACTCCTGCTT

## Data Availability

Data is contained within the article and Appendix A.

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
