# Peer review of "The Reparative Function of MMP13 in Tertiary Reactionary Dentinogenesis after Tooth Injury"

_ijms, 2024, doi:10.3390/ijms25020875_

Round 1

Reviewer 1 Report

Comments and Suggestions for Authors

The manuscript is entitled "The reparative function of MMP13 in tertiary reactionary dentinogenesis after tooth injury"

Overall, the manuscript provides a comprehensive investigation into the role of matrix metalloproteinase 13 (MMP13) in tertiary dentine formation after dental injury. The study is well-structured and employs a combination of in vivo models, histological analyses, and molecular assessments to examine the influence of MMP13 on odontoblast activity and dentine repair processes.

Here are some specific suggestions for improvement:

The manuscript is generally well-written, but some sentences are complex and may benefit from simplification for better readability.
It would be beneficial to discuss the potential limitations of the study.

The manuscript can be accepted in its present form.

Author Response

General comments: Overall, the manuscript provides a comprehensive investigation into the role of matrix metalloproteinase 13 (MMP13) in tertiary dentine formation after dental injury. The study is well-structured and employs a combination of in vivo models, histological analyses, and molecular assessments to examine the influence of MMP13 on odontoblast activity and dentine repair processes.

Response: Many thanks for your positive feedback

Comment 1: The manuscript is generally well-written, but some sentences are complex and may benefit from simplification for better readability.

Response 1: Thank you, in response, several sentences have been simplified for ease of reading.

Comment 2: It would be beneficial to discuss the potential limitations of the study.

Response 2: Thank you, the potential imitations of the study have now been added to the discussion: Line 446-451.

“This study is potentially limited by the fact that not does not analyse the comparative injury response after pulpal exposure, focusing only on the response to injury in the absence of exposure. Although reactionary dentinogenesis is of considerable importance the response of the exposed pulp is also of great translational interest and will be the focus of our future work in this area.”

Reviewer 2 Report

Comments and Suggestions for Authors

The reviewer appreciates the efforts of the authors to conduct this study which has good clinical significance. The study is well-designed to achieve the objectives. The manuscript is well written however, the reviewer noticed a few errors in the manuscript that need to be revised before acceptance.

Abstract:

Line 13: Typo of MMP 13

Line 14-16: “with research from our group previously demonstrating that global MMP13-deletion resulted in critical alterations in the dentine-phenotype, affecting dentine-tubule regularity, the odontoblast palisade and significantly reduced dentine volume.” The reviewer does admire the contribution of the author with previous publications on this topic however, it would be better to revise such a statement to be more scientifically sound

Line 20-21: “We demonstrated that MMP13 loss affected reactionary tertiary dentine quality…..” same as above

Introduction

Line 52-53: “Research from our group has shown that hydraulic calcium silicate cements (HCSCs) effectively promote reparative tertiary dentine formation” same as above

Line 77-78: “Recently our group has highlighted a key role for MMP13….” Same as above

Please add specific objectives of the study

If the author intends to test a hypothesis in the study, that should be properly structured and add acceptance or rejection of the same in the stating of the discussion section.

Result

The author has used different abbreviations (MMP and Mmp) for matrix metalloproteinase throughout the manuscript. Please use a uniform abbreviation.

Discussion

The current study evaluates and emphasizes the role of MMP 13 however, as per the result shown in Figure 8, MMP 8 and MMP 9 upregulated significantly higher than MMP 13. The author should add a comparative explanation in the discussion part that justifies the result.

Conclusion

Please add a subheading of the conclusion. The conclusion section should be specific to the outcome of the current study. Please avoid repetition of explaining facts stated in the discussion section and referring to previously published articles.

Materials & methods

The author has used different groups based on types of injury and time for each experiment. The reviewer highly recommends adding a flowchart diagram that summarizes the methodology section for a better understanding of the groups and variables used in each experiment.

Author Response

General comments: The reviewer appreciates the efforts of the authors to conduct this study which has good clinical significance. The study is well-designed to achieve the objectives. The manuscript is well written however, the reviewer noticed a few errors in the manuscript that need to be revised before acceptance.

Response: Thank you for the feedback, we have now modified the manuscript point-by-point as suggested by the reviewer.

Comment 1: Abstract: Line 13: Typo of MMP 13

Response 1: This has now been changed and for consistency, unless referring to gene expression specifically (where lower case italics are used as in convention) MMP13 is used throughout.

Comment 2: Abstract: Line 14-16: “with research from our group previously demonstrating that global MMP13-deletion resulted in critical alterations in the dentine-phenotype, affecting dentine-tubule regularity, the odontoblast palisade and significantly reduced dentine volume.” The reviewer does admire the contribution of the author with previous publications on this topic however, it would be better to revise such a statement to be more scientifically sound

Response 2: In order to make more scientifically sound, the personal pronouns have been removed as indicated.

Comment 3: Abstract: Line 20-21: “We demonstrated that MMP13 loss affected reactionary tertiary dentine quality…..” same as above

Response 3: Thanks, changed now to “MMP13 loss affected reactionary tertiary dentine quality…..”

Comment 4: Introduction. Line 52-53: “Research from our group has shown that hydraulic calcium silicate cements (HCSCs) effectively promote reparative tertiary dentine formation” same as above

Response 4: This has now been changed to “Previous research has shown that hydraulic calcium silicate cements (HCSCs) effec-tively promote reparative tertiary dentine formation”

Comment 5: Introduction. Line 77-78: “Recently our group has highlighted a key role for MMP13….” Same as above

Response 5: Thank you, in line with this comment the sentence has been modified to “Recently a key role for MMP13 in mineralization in osteoblasts [31], dental pulp cells (DPC) [32] and dentine formation [20] has been highlighted.”

Comment 6: Please add specific objectives of the study

Response 6: Thank you, specific objectives have now been added at the end of the introduction.

“The specific objectives are to analyse qualitatively and quantitively the nature and volume of tertiary dentine deposition in mouse molars after cuspal injury as well as to assess mechanistic links to the Wnt-signalling pathway.”

Comment 7: If the author intends to test a hypothesis in the study, that should be properly structured and add acceptance or rejection of the same in the stating of the discussion section.

Response 7: As specific objectives (as was requested above) have now been added the hypothesis element has been removed and aims and objectives expanded.

Comment 8: Results: The author has used different abbreviations (MMP and Mmp) for matrix metalloproteinase throughout the manuscript. Please use a uniform abbreviation.

Response 8: Thank you. MMP is now used throughout, except for genes expression, when the scientifically accepted nomenclature of Mmp13 or Mmp13-/- is used.

Comment 9: Discussion: The current study evaluates and emphasizes the role of MMP 13 however, as per the result shown in Figure 8, MMP 8 and MMP 9 upregulated significantly higher than MMP 13. The author should add a comparative explanation in the discussion part that justifies the result.

Response 9: This is a good point, MMP8 and 9 are classic proinflammatory markers and would be expected to increase after acute injury and contamination. MMP13’s role after inflammation is unclear, as previous research has focussed on mineralisation effects. In response to the comment text has been added to the discussion to make this clearer.

“Classic markers of pulpitis, MMP9 [25] and MMP8 [37] were as expected significantly increased after pulp exposure and contamination with oral bacteria; however, MMP13 also showed a significant increase in expression (approximately 1.6-fold increase) of a similar magnitude to MMP8, but not as great as MMP9 (approximately 6-fold increase), which highlights a potentially novel role for MMP13 in inflammation as well as repair within the dentine-pulp complex.”

Comment 10: Conclusion. Please add a subheading of the conclusion. The conclusion section should be specific to the outcome of the current study. Please avoid repetition of explaining facts stated in the discussion section and referring to previously published articles.

Response 10: Conclusion now a subheading and all references to previous work removed as requested.

Comment 11: Materials & methods. The author has used different groups based on types of injury and time for each experiment. The reviewer highly recommends adding a flowchart diagram that summarizes the methodology section for a better understanding of the groups and variables used in each experiment.

Response 11: Thank you. A flow chart has now been added as Supplementary Figure 1. It would be uncommon to see such a flowchart in the main manuscript of biological papers.